# Sex and HDAC4 Differently Affect the Pathophysiology of Amyotrophic Lateral Sclerosis in SOD1-G93A Mice

**DOI:** 10.3390/ijms24010098

**Published:** 2022-12-21

**Authors:** Alessandra Renzini, Eva Pigna, Marco Rocchi, Alessia Cedola, Giuseppe Gigli, Viviana Moresi, Dario Coletti

**Affiliations:** 1DAHFMO Unit of Histology and Medical Embryology, Sapienza University of Rome, 00161 Rome, Italy; 2Department of Biomolecular Sciences, University of Urbino Carlo Bo, 61029 Urbino, Italy; 3Institute of Nanotechnology, c/o Dipartimento di Fisica, National Research Council (CNR-NANOTEC), Sapienza University of Rome, 00185 Rome, Italy; 4Institute of Nanotechnology, c/o Campus Ecotekne, National Research Council (CNR-NANOTEC), Monteroni, 73100 Lecce, Italy; 5Biological Adaptation and Ageing, Institut de Biologie Paris-Seine, Sorbonne Université, F-75005 Paris, France

**Keywords:** ALS, neurogenic muscle atrophy, velocity of weight loss, transgenic mice

## Abstract

Amyotrophic Lateral Sclerosis (ALS) is a devastating adult-onset neurodegenerative disease, with ineffective therapeutic options. ALS incidence and prevalence depend on the sex of the patient. Histone deacetylase 4 (HDAC4) expression in skeletal muscle directly correlates with the progression of ALS, pointing to the use of HDAC4 inhibitors for its treatment. Contrarily, we have found that deletion of HDAC4 in skeletal muscle worsened the pathological features of ALS, accelerating and exacerbating skeletal muscle loss and negatively affecting muscle innervations in male SOD1-G93A (SOD1) mice. In the present work, we compared SOD1 mice of both sexes with the aim to characterize ALS onset and progression as a function of sex differences. We found a global sex-dependent effects on disease onset and mouse lifespan. We further investigated the role of HDAC4 in SOD1 females with a genetic approach, and discovered morpho-functional effects on skeletal muscle, even in the early phase of the diseases. The deletion of HDAC4 decreased muscle function and exacerbated muscle atrophy in SOD1 females, and had an even more dramatic effect in males. Therefore, the two sexes must be considered separately when studying ALS.

## 1. Introduction

Amyotrophic Lateral Sclerosis (ALS) is a devastating motor neuron disease primarily caused by a progressive loss of upper and lower motor neurons that leads to muscular atrophy, paralysis, and premature death. ALS exists in two different forms: the sporadic form, with no apparent inheritability, and the familial form. Approximately 15–20% of the familial ALS cases and about 5% of the sporadic ALS cases are associated with a genetic form, due to multiple mutations in the gene SOD1, superoxide dismutase 1 [1]. Furthermore, individual factors contribute to determining the pronounced clinical heterogeneity of the disease. Among these factors, sex differences are observed, as detailed below. 

ALS incidence and prevalence are about 1.3 times higher in men than in women, with a higher number of men experiencing ALS diagnosis earlier than women, quantifiable as an onset that is 5 years earlier [2]. Differences in clinical phenotypes have also been reported between male and female patients, especially regarding the site of onset, limb or bulbar, and the involvement of cognitive impairment [2,3,4]. Sex differences have been described both in sporadic and familial ALS forms [4]. In addition, ALS in men is more often diagnosed as a sporadic form than in women [5]. Sex-based immunological [6,7], hormonal [8], and metabolic [9] variances underpin differences between females and males in ALS onset, progression, and prognosis.

Sex differences in ALS have also been reported in genetic mouse and rat models, which are obtained by overexpressing the human *SOD1* gene harboring a glycine to alanine mutation at amino acid 93 (hSOD1-G93A) [10,11,12]. As for the SOD1-G93A (SOD1) mouse phenotype, it closely recapitulates the human clinical and histopathological symptoms of ALS, despite the relatively early onset and rapid progression of the disease [13]. Even though SOD1 mice have been extensively used for studying the mechanisms of the pathology, as well as the effectiveness of drug treatment strategies, to date, conflicting reports on the effect of sex on ALS onset and mouse lifespan have been reported. Some studies describe a longer lifespan, in addition to delayed onset, in female mice compared to male mice [10,11], while others find no difference in the disease endpoint between sexes [14,15,16]. Such discrepancies represent a scientific gap that needs to be fulfilled prior to assessing any sex-related differences in therapeutic treatments or in the biological functions of a specific factor.

Histone Deacetylase 4 (HDAC4) is a member of class IIa HDACs, whose expression in skeletal muscle is significantly higher in the ALS patients who experience a more rapid progression of the disease. It negatively correlates with the extent of muscle reinnervation and muscle function [17,18]. By generating SOD1 mice with a skeletal muscle-specific deletion of HDAC4 (namely SOD1 HDAC4mKO mice), we recently demonstrated that, in male mice, deletion of HDAC4 in the skeletal muscle worsens the pathological features of ALS, advancing and exacerbating both body weight and skeletal muscle loss, in addition to negatively affecting neuromuscular junction integrity and muscle innervations [19].

Since it is not yet clear whether there is a sex-dependent effect on ALS onset and progression in SOD1 mice, nor whether HDAC4 exerts different sex-related functions in ALS muscles, in the present work, we aimed to investigate: (1) ALS outcome in terms of body weight changes, disease onset, and mouse lifespan, comparing the two sexes; and (2) a previously unreported role for HDAC4 function in the skeletal muscle of SOD1 female mice.

## 2. Results

### 2.1. Sex and HDAC4 Influence Body Weight in SOD1 Mice

A rapid body weight decline is associated with a shorter survival time in ALS patients. Three independent studies, on a total of almost 800 ALS patients, concluded that the rapidity of body weight loss in the initial stages of the disease is a strong and independent prognostic factor in ALS, while weight gain after diagnosis improves survival, especially in female patients [20,21,22]. Thus, we relied on the body weight changes to assess ALS onset and progression.

We analyzed the kinetics of body weight and found that SOD1 male mice weighed significantly more than female mice (Figure 1a). To shed light on the role of HDAC4 in the progression and output of ALS, we generated a SOD1 HDAC4mKO mouse line by breeding SOD1 mice with mice harboring a skeletal muscle-specific deletion of HDAC4, as previously described [19]. We analyzed the kinetics of body weight, confirming that SOD1 HDAC4mKO male mice weighed significantly more than female mice. In addition, time (i.e., the age of the mouse) had a significant effect on the body weight, which declined with the progression of the disease (Figure 1a).

The differences in body weight, and the interaction between time and genotype, suggested that these variables should be carefully considered. Therefore, we normalized the body weight of each mouse to its body weight at 10 weeks of age, to eliminate bias factors such as the sex-related differences in the initial body weight. With this approach, we could better delineate the effects of sex and genotype on the kinetics of body weight loss. In particular, sex-dependent differences in normalized body weight were detected from week 16 to 18 in SOD1 mice, because females continued to gain body weight until 17 weeks, a phenomenon that was not observed in males (see * values in Figure 1b).

The deletion of HDAC4 negatively affected body weight in both sexes. However, female SOD1 HDAC4mKO mice significantly lost body weight at 19 weeks, as compared to their body weight at 10 weeks, while male SOD1 HDAC4mKO mice started to lose weight earlier, i.e., at 17 weeks (see # in Figure 1b). Moreover, in females, the deletion of HDAC4 hampered the weight gain previously described, resulting in a normalized weight significantly lower in SOD1 HDAC4mKO females compared with that of SOD1 females from week 16 to 18 (see pink $ in Figure 1b). In males, the deletion of HDAC4 induced a more rapid drop in body weight, causing the normalized weight of SOD1 HDAC4mKO males to be significantly lower than that of SOD1 males from week 17 to 19 (see blue $ in Figure 1b). The different kinetics of body weight changes and its differential severity suggest striking differences between the two sexes in the development of ALS.

### 2.2. Sex and HDAC4 Differentially Affect ALS Onset and Survival in SOD1 Mice

The onset is of particular relevance for progressive, degenerative diseases, such as ALS. Based on all of the above, the disease onset was retrospectively defined as the time when body mass starts declining. Given the complex kinetics of body weight changes described above, we decided to monitor ALS onset, plotting the mice in Kaplan–Meier curves and performing a global analysis by using the Cox regression test. With this approach, a significant effect on ALS onset was detected for the two variables, i.e., sex (*p* = 0.008) and genotype (*p* = 0.002) (Figure 2a). Importantly, male mice showed a hazard ratio to develop ALS which was 2.4 times higher than that of females, while deletion of HDAC4 in SOD1 skeletal muscles raised the hazard ratio of ALS onset by 2.3 times compared to SOD1 muscles expressing HDAC4. When narrowing the analysis to the comparison of the male and female subpopulations, we found no significant differences (*p* = 0.148) between SOD1 females and males, nor between SOD1 HDAC4mKO females and males (*p* = 0.115), suggesting that sex had a minor role in the onset of the disease in SOD1 mice, which only emerges in global analysis. Instead, when the comparison was carried out to specifically address the role of HDAC4, i.e., comparing SOD1 and SOD1 HDAC4mKO males, significant differences were detected (*p* = 0.01), confirming previous results [19]. Finally, no significant differences between SOD1 and SOD1 HDAC4mKO females (*p* = 0.096) were found. Altogether, these observations indicate that the deletion of HDAC4 has a more dramatic effect on the onset of ALS in male mice than in female mice.

The Cox global analysis, which was applied to the data of ALS progression and is defined as the time from the onset of the disease to the death of the animal, revealed that no differences are detectable as due to sex or lack of HDAC4 (Appendix A). Therefore, based on the overlapping curves and considering that once the disease is overt, its progression is similar between males and females in the absence or presence of HDAC4, we did not proceed to compare the four experimental groups.

As for the survival, we observed highly significant differences depending on the sex of the animals (*p* = 0.001) (Figure 2b). By comparing the groups, no significant sex-dependent differences were revealed in SOD1 mice (*p* = 0.153); instead, highly significant differences were detected between females and males in SOD1 HDAC4mKO mice (*p* < 0.0001). While no significant differences were detected between SOD1 and SOD1 HDAC4mKO females (*p* = 0.3), significant differences occurred between SOD1 and SOD1 HDAC4mKO males (*p* = 0.001) (Figure 2b), further confirming that the deletion of HDAC4 has a more dramatic effect in male than in female SOD1 mice.

Given the relevance of lifespan, we focused on the potential correlation between the extension of survival and the time of disease onset, as well as the velocity of body weight loss. To do so, we calculated the exponential regression, correlating the lifespan with the time of ALS onset for each individual mouse. We found a significant positive correlation, indicating that the later ALS onset occurs, the longer the survival of the mouse (Figure 2c). Then, we calculated the velocity of daily body weight loss between 18 and 17 weeks of age, since this is the age at which the onset of the disease becomes manifest in all the mice (i.e., 17 weeks), but the survival is still high and most of the animals are alive (i.e., 18 weeks). We then correlated the lifespan with the velocity of body weight loss of the corresponding mouse. A significant inverse correlation between the survival time and the velocity of weight loss was found, indicating that faster body weight loss is associated with a shorter survival time of the mouse (Figure 2d).

In conclusion, by comparing female and male mice, we noticed a global effect of sex on ALS onset and survival, with males exhibiting earlier onset and dying earlier than female mice. The deletion of HDAC4 in skeletal muscle anticipates the onset of the disease and death in SOD1 male mice, suggesting that these mice die earlier since the disease appears earlier. All of the above indicate that the deletion of HDAC4 is more detrimental in male SOD1 mice than in female SOD1 mice.

### 2.3. Deletion of HDAC4 Worsens the Pathological Features of Female SOD1 Muscles

With the aim to characterize the functions of HDAC4 in the skeletal muscle of female mice, we analyzed the female phenotype of the pathology in the absence or presence of HDAC4, since this was still lacking a characterization. In particular, we investigated muscle function, mass, and morphology at a pre-onset stage (i.e., 12 weeks of age) and at a late stage (i.e., 18 weeks of age) of the disease. To assess whether HDAC4 affects muscle strength, a grip test was performed in SOD1 and SOD1 HDAC4mKO female littermates. Importantly, SOD1 HDAC4mKO mice showed a significant reduction in muscle force as early as at 12 weeks of age compared to SOD1 littermates (Figure 3a), despite no significant differences in the Tibial Anterior (TA) or Gastrocnemius (GA) muscle mass and TA histology (Figure 3b–d).

At a late stage of the disease, in addition to the decrease in muscle force (Figure 4a), significant differences were observed in TA and GA muscle weights, as well as in TA muscle morphology, between SOD1 and SOD1 HDAC4mKO female littermates (Figure 4b–d). Indeed, morphometric analyses by using the minimum Feret’s diameter confirmed a more pronounced muscle atrophy in SOD1 HDAC4mKO muscle fibers, with a shift in myofiber size toward a smaller size (0-30 μm), compared to SOD1 female littermates (Figure 4d).

All of the above confirm that deletion of HDAC4 ultimately affects female SOD1 muscles. Importantly, muscle dysfunction emerges as an early sign of the ALS disease in skeletal muscle, in the absence of other pathological features.

## 3. Discussion

The importance of sex differences in medicine is now widely acknowledged: for disease states as different as fungal infection and cardiomyopathy, it is evident that differences between males and females are significant in terms of genetic bases and susceptibility, disease progression, and survival for the most severe pathologies [23,24,25]. Therefore, diagnosis, therapy, and palliative care must be tailored to the patient, or, minimally, must consider the sex of the patient. Our study stems from contradictory data currently present in the literature regarding possible sex-related differences in ALS onset and lifespan of SOD1 mice, one of the most widely used experimental models for studying ALS in vivo. Establishing whether sex affects pathological features of ALS in SOD1 mice is a prerequisite to defining the role of specific biological factors or the effectiveness of any therapeutic approaches. Indeed, discovering how a specific factor or therapy affects each sex will help in the development of personalized, patient-specific therapies.

The general issue of body mass and its impact on disease progression is highly debated for many different diseases, and is well-represented by the obesity paradox: while an excess of body weight has been considered a risk factor for health, it may result in a protective effect in many disease conditions after the onset of the pathology (reviewed by Bosello [26]). For instance, in the case of cancer cachexia, a syndrome primarily characterized by body weight loss [27], hypercholesterolemia and obesity are paradoxically associated with better survival [28]. In ALS, the velocity of weight loss, independently from the initial body weight, is a clear independent prognosis factor which directly correlates with the risk of death of patients [22,29]. In our opinion, this suggests that the kinetics of body weight is a more accurate way to analyze a progressive phenomenon, such as ALS, and a more informative parameter to be considered for studying this disease; indeed, body weight changes are the global result of a complex response to a disease. The issue of body weight shall be analyzed in its kinetics, i.e., in a dynamic way, to provide informative data related to both diagnosis and prognosis. By analyzing raw body weights over time, expected, gross differences between the two sexes were observed, male mice being heavier than female ones: this occurs in healthy mice and holds true for the transgenic mice used in this study. To unveil differences among the ALS males and females, in the absence or presence of HDAC4, and to eliminate the sex-related differences in the initial body weight, we normalized the body weight of each mouse relative to a starting point. Interestingly, SOD1 female mice continued to gain body weight until week 17 of age; instead, differently from healthy mice [30], SOD1 male mice remained stable in their body weight. Considering the kinetics and the higher body weight of male mice, it is not directly predictable whether males would be prone to a severe form of the disease. Notably, deletion of HDAC4 negatively affected body weight kinetics in both sexes: while, in females, it hampered the weight gain, in males, it induced a more rapid drop in body weight.

In humans, it should be considered that ALS onset is a totally subjective parameter and may be not as accurate as in mice. Notably, different conclusions on the onset of the disease in mice have already been reported in the literature, depending on the test applied. Sex differences in ALS onset have also been reported [15,16], by calculating the onset with different specific tests (hanging wire, rotarod, and electromyography) or hind paw extension reflex, respectively. Even by taking into account the same type of parameters, such as tremor onset, contradictory data have been reported on sex differences; some authors reported a delayed clinical manifestation in female SOD1 mice [16,31], while no sex-related differences exist for others [14,32]. Such discrepancies may be due to the subjective nature of the parameter chosen, i.e., initial tremor. In order to choose an objective parameter, we took into consideration the time of maximum body weight, and we postulated that the onset of the disease occurs at this maximum point of the curve, i.e., when the mice start losing weight. When analyzing the global effect of sex on ALS onset (defined as the beginning of body weight loss) or survival, we did observe a significant effect of sex on these variables, with females showing a later onset and prolonged survival as compared to males, as in humans [2]. We found that male mice have a hazard ratio to develop ALS that is 2.4 times higher than that of female mice, while deletion of HDAC4 in SOD1 skeletal muscles raises the hazard ratio of ALS onset by 2.3 times compared to SOD1 muscles expressing HDAC4. If we compared the four experimental groups separately, though, no significant differences between sexes were detected in ALS onset, similar to another study [14].

In the search for the major contributor to the survival of SOD1 mice, we correlated the age of onset and the velocity of weight loss with survival. We found that both variables correlate with survival, in opposite ways. Indeed, the higher the age of onset, the longer the survival, while the more rapidly weight loss occurs, the shorter the survival. These conclusions are in agreement with the data on human patients, since the age of ALS onset in patients [33] and velocity of the decline in body weight [22,29] are considered prognostic factors, while possessing an increased body mass is not protective against muscle wasting or premature death, per se.

HDAC4 deletion in the skeletal muscle significantly accelerates the onset of ALS in male SOD1 mice, thus resulting in a significant decrease in their survival time. In female SOD1 mice, instead, HDAC4 deletion did not apparently affect the age of onset or the time of survival. This data seemed to be in contrast with the observed negative effects of HDAC4 deletion on the body weight of female mice. Thus, we investigated muscle atrophy in terms of muscle strength, mass, and morphology in female SOD1 and SOD1 HDAC4mKO mice, at a pre-symptomatic and late stage of the disease. We demonstrated that the deletion of HDAC4 in the skeletal muscle worsens muscle function in SOD1 HDAC4mKO female mice even before the ALS onset, as in male mice [19], and exacerbates the ALS pathological features at a late stage of the disease, confirming the protective role of HDAC4 in skeletal muscle in ALS. Previously, we have reported the crucial importance of HDAC4 in Duchenne Muscular Dystrophy [34], which is consistent with its protective role reported for another disease. Considering the pivotal functions that HDAC4 plays in skeletal muscle response to denervation [35], and the role in ALS shown here and recently characterized elsewhere [19], HDAC4 emerges as a major player in granting the preservation of muscle integrity and function.

Our results indicate that HDAC4 exerts a protective role in ALS in both sexes, still highlighting sex-related differences in this disease. Our observations should be considered for the use of Histone Deacetylase inhibitors in therapy for ALS, since its inhibition may be detrimental to skeletal muscle regardless of the sex of the patient.

## 4. Materials and Methods

### 4.1. Mice

B6SJL-Tg(SOD1*G93A)1Gur/J (Common Name: SOD1-G93A) transgenic mice (Charles River Laboratories) were crossed with Hdac4^fl/fl^ myogenin;Cre mice (generously provided by Eric N. Olson) to obtain SOD1-G93A Hdac4fl/fl mice (mice with ALS disease and homozygous for a floxed Hdac4 allele, referred to as SOD1 mice) and SOD1-G93A Hdac4fl/fl myogenin;Cre mice (mice with ALS disease and KO for HDAC4 upon myogenin expression, referred to as SOD1 HDAC4mKO mice). Mice were treated in strict accordance with the guidelines of the Institutional Animal Care and Use Committee, as well as national and European legislation, throughout the experiments. Animal protocols were approved by the Italian Ministry of Health (authorizations # 244/2013 and # 853/2016-PR). To determine the extent of sex-related differences in ALS, male and female SOD1 mice from the same litter were compared, while to define HDAC4 functions in the skeletal muscle in ALS, male or female SOD1 HDAC4mKO and SOD1 mice from the same litter were compared. To determine body weight over time, each mouse was weighed three times per week and the mean of the measurements was calculated. The age of ALS onset (day and week) was determined as the time when the maximum body weight was achieved. As for the survival, the experiment was ended when mouse paralysis was so severe that the animal could not right itself.

### 4.2. Functional Analyses

To measure muscle strength, grip test analyses were performed by holding mice by the tail and allowing them to grab a T-shaped bar connected to a force transducer, in turn connected to the peak amplifier 47105–001 (Ugo Basile, Gemonio (VA), Italy). Mice were then pulled backward, exerting a constant force until they lost the grip. The force applied to the bar was recorded as the peak tension. To obtain reproducible data, each animal was subjected to 10 consecutive measurements, and the mean values were used.

### 4.3. Histological Analyses

The TA muscles from female SOD1 and SOD1 HDAC4mKO littermates were analyzed. Muscles were dissected, embedded in tissue freezing medium (Leica, Wetzlar, Germany), and frozen in liquid nitrogen pre-cooled isopentane (Sigma-Aldrich, Saint Louis, Missouri, USA, # PHR1661). Cryosections (8 μm) were obtained by using a Leica cryostat (Leica, Wetzlar, Germany).

Hematoxylin and eosin (H&E) (Sigma-Aldrich, Saint Louis, MO, USA, # H3136 and # 861006) staining was performed according to the Sigma-Aldrich manufacturer’s instructions.

### 4.4. Morphometric Analyses

The minimum Feret’s diameter was quantified on the TA cryosections stained with H&E by using Image J software (NIH, Bethesda, MA, USA). The entire muscle section was quantified for each biological replicate.

### 4.5. Statistics

The differences between the sexes and genotypes were confirmed by independent experiments, i.e., independent litters, by considering as “n” an independent biological sample, not a technical replicate. After verifying the normal distribution of the data and their homoscedasticity by means of Levene’s test, data were analyzed with two-way repeated measures ANOVA, followed by Tukey’s HSD as post hoc test. Student’s *t*-test was used to compare the two groups. All values were expressed as mean ± standard error of the mean (SEM). A Cox regression was used to investigate the effect of sex and genotype, on the time of onset, progression, and survival. The specific test used, the global effects, the specific post hoc test used, and the significance level of specific comparisons are indicated in the figure legends, which also indicate the size of the samples analyzed. All the statistical analyses and graphs were conducted by using Prism software (GraphPad by Dotmatics, San Diego, CA, USA).

## 5. Conclusions

Sex differences need to be taken in consideration when studying ALS in the murine SOD1 mouse model. HDAC4 has a significant role in this neurodegenerative disease, affecting the kinetics of body weight, muscle function, and atrophy, in both male and female mice.

## Figures and Tables

**Figure 1 ijms-24-00098-f001:**
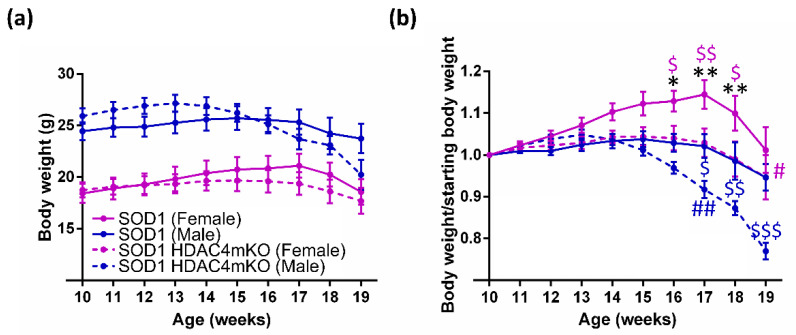
Sex and HDAC4 influence body weight in SOD1 mice. (**a**) Body weight of female and male SOD1 and SOD1 HDAC4mKO mice. Data are expressed as average +/- SEM; n = 10/13 females and n = 14 males for each time point. Two-way repeated measures ANOVA reveals a significant effect for time (*p* < 0.0001), genotype (*p* < 0.0001), and a significant interaction (*p* < 0.0001); *p* < 0.005 female SOD1 vs. male SOD1; *p* = 0.0001 female SOD1 HDAC4mKO vs. male SOD1 HDAC4mKO mice, by Tukey’s multiple comparisons test. (**b**) Body weight of female and male SOD1 and SOD1 HDAC4mKO mice, normalized to the body weight at 10 weeks. Data are expressed as average +/- SEM; n = 10/13 females and n = 14 males for each time point. Two-way repeated measures ANOVA revealed a significant effect for time (*p* < 0.0001) and for genotype (*p* = 0.001), and a significant interaction (*p* < 0.0001): $ *p* < 0.05, $$ *p* < 0.005, $$$ *p* < 0.0001. Male SOD1 vs. male SOD1 HDAC4mKO mice, as well as female SOD1 vs. female SOD1 HDAC4mKO mice: * *p* < 0.05, ** *p* < 0.005. Female SOD1 vs. male SOD1 mice: # *p* < 0.05, ## *p* < 0.005. Female SOD1 HDAC4mKO vs. their body weight at 10 weeks, or male SOD1 HDAC4mKO vs. their body weight at 10 weeks, were evaluated by Tukey’s multiple comparisons test.

**Figure 2 ijms-24-00098-f002:**
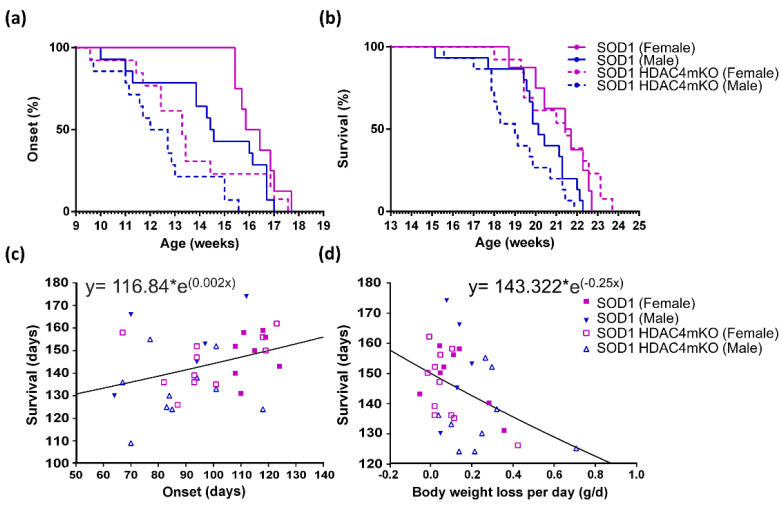
Sex and HDAC4 differentially affect the onset and lifespan of female and male SOD1 mice. (**a**) Age of ALS onset of female and male SOD1 and SOD1 HDAC4mKO mice. n = 12 females and n = 14 males for each genotype. *p* = 0.001 for global effect, *p* = 0.008 for sex, and *p* = 0.002 for genotype by Cox regression test. (**b**) Survival curve of SOD1 and SOD1 HDAC4mKO mice. n = 8 females and n = 14 males for each genotype; *p* = 0.001 for sex by Cox regression test. (**c**) Exponential curve correlating the lifespan and the time of ALS onset in female and male SOD1 and SOD1 HDAC4mKO mice. n = 34; R square = 0.131; *p* = 0.035. (**d**) Exponential curve correlating the lifespan and the velocity of body weight loss (g per day) in female and male SOD1 and SOD1 HDAC4mKO mice. n = 33, R square = 0.178; *p* = 0.015.

**Figure 3 ijms-24-00098-f003:**
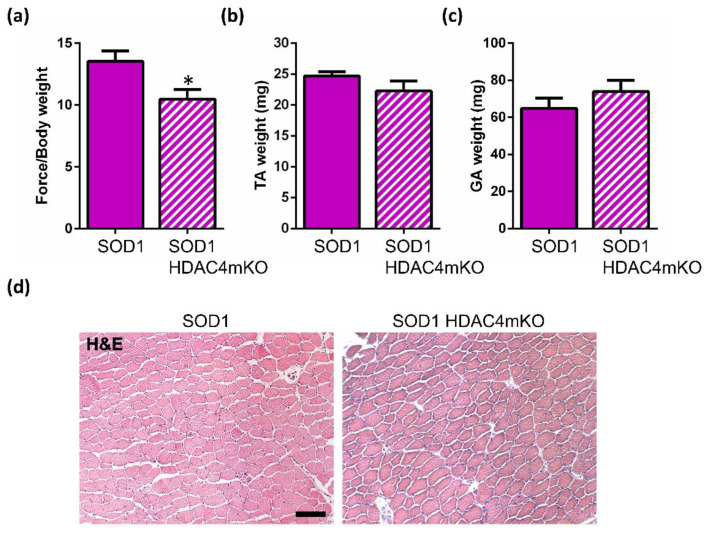
Deletion of HDAC4 negatively affects muscle performance in female SOD1 mice at a pre-symptomatic stage of the disease. (**a**) Muscle performance of SOD1 and SOD1 HDAC4mKO mice by grip test, at 12 weeks of age. Data are expressed as the mean of developed force over the mouse weight, +/− SEM. N = 3 mice for genotype; * *p* < 0.05 by Student’s *t*-test. (**b**) Tibialis Anterior (TA) muscle weight of SOD1 and SOD1 HDAC4mKO mice at 12 weeks of age. Data are expressed as mean +/− SEM. n = 5 SOD1 and n = 7 SOD1 HDAC4mKO mice. (**c**) Gastrocnemius (GA) muscle weight of SOD1 and SOD1 HDAC4mKO mice at 12 weeks of age. Data are expressed as mean +/− SEM. n = 8 SOD1 and n = 10 SOD1 HDAC4mKO mice. (**d**) Representative images of TA muscle histology of female SOD1 and SOD1 HDAC4mKO littermates, at 12 weeks of age. Scale bar = 100 μm.

**Figure 4 ijms-24-00098-f004:**
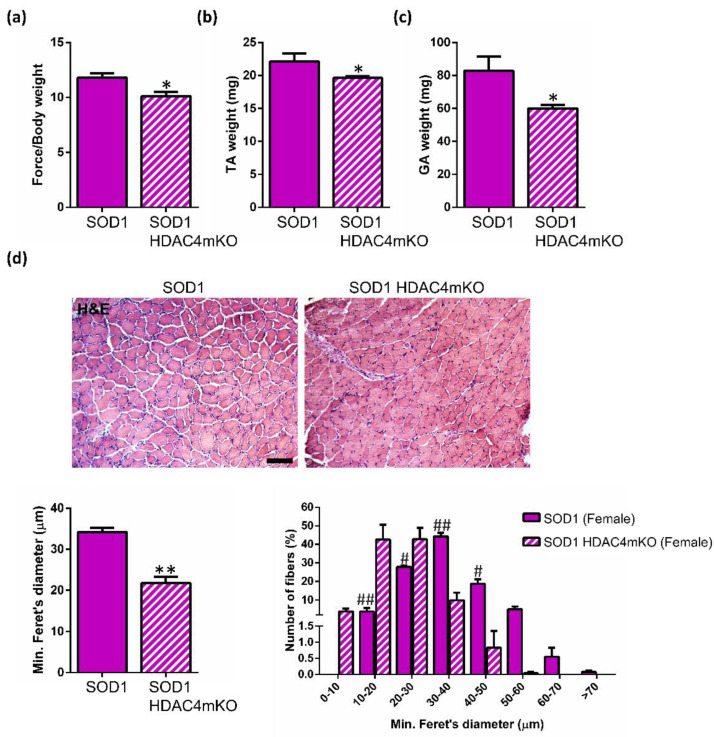
Deletion of HDAC4 worsens muscle atrophy in female SOD1 mice. (**a**) Muscle performance of SOD1 and SOD1 HDAC4mKO mice by grip test, at 18 weeks of age. Data are expressed as the mean of developed force over the mouse weight, +/− SEM. n = 4 mice for genotype; * *p* < 0.05 by Student’s *t*-test. (**b**) TA muscle weight of SOD1 and SOD1 HDAC4mKO mice at 18 weeks of age. Data are expressed as mean +/− SEM. n = 5 SOD1 and n = 7 SOD1 HDAC4mKO mice; * *p* < 0.05 by Student’s *t*-test. (**c**) GA muscle weight of SOD1 and SOD1 HDAC4mKO mice at 18 weeks of age. Data are expressed as mean +/− SEM. n = 3 mice for each genotype; * *p* < 0.05 by Student’s *t*-test. (**d**) Representative images of TA muscle histology of female SOD1 and SOD1 HDAC4mKO littermates, at 18 weeks of age. Scale bar = 100 μm. Minimum Feret’s diameter of SOD1 and SOD1 HDAC4mKO TA myofibers at 18 weeks of age. Data are expressed as mean +/− SEM. Myofiber minimum Feret’s diameter distribution. Data are expressed as mean +/− SEM. n = 3 mice for each genotype; ** *p* < 0.005, # <0.001, ## <0.0001 by Student’s *t*-test.

## Data Availability

Not applicable.

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
