# Peer review of "Sex and HDAC4 Differently Affect the Pathophysiology of Amyotrophic Lateral Sclerosis in SOD1-G93A Mice"

_ijms, 2022, doi:10.3390/ijms24010098_

Round 1
Reviewer 1 Report (Previous Reviewer 2)
This manuscript represents an incremental advance on their previously published work with new incorporation of un-normalised weight data and muscle histology/functional readout results for female mice. A greater focus on comparing females to males in this respect would be best considering the males are the sex showing notable changes regarding weight and survival.
Overall, it comes as no surprise that weight differences are observed as HDAC4 knocked out of muscles coupled with denervation by mutant SOD1 will exacerbate weight loss through muscular atrophy. This also explains the onset changes which the author's state are dictated by the point at which peak weight was observed. If anything, these results in the females suggest that HDAC4 plays a role in supporting muscular function by has no effect on neuroprotection represented through survival. This would ideally be done through doing parallel motor neuron counts to confirm a lack of change on this side of the muscle-nerve axis and provide a more balanced discussion addressing this aspect.
Aside from these strong recommendations, I have a few points:
1) Spelling mistakes are still present (e.g., “breading” instead of “breeding” in line 81) and should be proof-read.
2) While the use of the word “risk” is understood in the context of Cox regression analyses, it makes little sense that “male mice have a 2.4 higher risk to develop ALS…”. They’re all over-expressing mutant SOD1G93A and as such have a 100% chance of developing their ALS phenotype. Stating these results in the form of “hazard ratio” would eliminate this paradoxical interpretation.
Author Response
This manuscript represents an incremental advance on their previously published work with new incorporation of un-normalised weight data and muscle histology/functional readout results for female mice.
We thank the reviewer for major and minor remarks, that, we believe, contributed to the amelioration of the manuscript.
A greater focus on comparing females to males in this respect would be best considering the males are the sex showing notable changes regarding weight and survival.
The reviewer 2 raised similar concerns, so please, refer to the response to reviewer 2 on this issue, too.
In keeping with the need to focus on female-to-male comparison, we have separated the discussion of the data first by comparing females to males SOD1 mice, and then comparing the phenotypes in the absence or presence of HDAC4, as much as possible. For instance, see lines 84 (sex) and 89 (genotypes) for the body weight. To further help the legibility of the results, we referred in the text to statistic symbols when describing specific, significant differences (see line 99 as an example).
The statistical approaches we choose (2-way ANOVA, Cox etc.) actually take into account the two variables, i.e. sex and genotype, together by definition. For instance, when we described the significant effect on ALS onset for the two variables, i.e. sex (p=0.008) and genotype (p=0.002), by using the Cox regression test (see line 117), within the variable “sex” both genotypes are present, and vice versa. Based on all the above, the disjoint discussion of the sex differences and those due to the genotype is rarely possible.
Indeed, we think that considering the two variables together represents the major innovation and advance of our work; consequently, we chose not to change the analyses and statistics performed, but rather to implement the description of the results in the text.
Overall, it comes as no surprise that weight differences are observed as HDAC4 knocked out of muscles coupled with denervation by mutant SOD1 will exacerbate weight loss through muscular atrophy. This also explains the onset changes which the author's state are dictated by the point at which peak weight was observed. If anything, these results in the females suggest that HDAC4 plays a role in supporting muscular function by has no effect on neuroprotection represented through survival. This would ideally be done through doing parallel motor neuron counts to confirm a lack of change on this side of the muscle-nerve axis and provide a more balanced discussion addressing this aspect.
We have previously demonstrated in males that the deletion of HDAC4 in the skeletal muscle of SOD1 mice does not affect motor neuron number, even if it advances mouse death (EBioMedicine. 2019 Feb;40:717-732. doi: 10.1016/j.ebiom.2019.01.038) and leads to a worse phenotype if compared to female SOD1 HDAC4mKO mice. Based on this evidence, we did not collect spinal cords from female mice, and we do not have the possibility to perform the analysis of motoneurons.
Aside from these strong recommendations, I have a few points:
1) Spelling mistakes are still present (e.g., “breading” instead of “breeding” in line 81) and should be proof-read.
We have carefully reviewed the text and fixed them.
2) While the use of the word “risk” is understood in the context of Cox regression analyses, it makes little sense that “male mice have a 2.4 higher risk to develop ALS…”. They’re all over-expressing mutant SOD1G93A and as such have a 100% chance of developing their ALS phenotype. Stating these results in the form of “hazard ratio” would eliminate this paradoxical interpretation.
We have replaced the word “risk” with “hazard ratio”, as suggested.

Reviewer 2 Report (New Reviewer)
The authors explore 2 sets of research questions. The first concern whether there are sex dependent differences among SOD1-G93A transgenic mice concerning 1) the age at which maximum weight is achieved (pitched as a proxy for disease onset), 2) the rate of weight loss subsequent this timepoints and 3) total lifespan. The second set of questions concern whether there are sex specific differences in the phenotypic consequences of HDAC4 deletion in G93A mice. This question is relevant because HDAC4 inhibition has been proposed as a therapeutic strategy in ALS, but the authors have previously reported that HDAC4 deletion exacerbated negative outcomes in male G93A mice.
Sex specific effects are an important focus in ALS research, with several unresolved controversies in the literature. The authors do observe some phenotypic differences across sexes in their study, however I think the results could be presented a little more clearly and that certain data would perhaps fit better as supplemental materials. I think the extent of the conclusions that can be made for sex dependent differences of SOD1-G93A genotypes has some important limitations. I think the conclusion that the phenotypic impact HDAC4 manipulation in ALS contexts shows some variation across sex seems valid, but the authors could elaborate on why they feel this is important a little more in the discussion.
Additional specific comments:
* I found the title misleading. I read it as the authors would propose that HDAC4 was a mediator of sex dependent differences in the phenotype of G93A mice. Perhaps the take home message could be clearer
* The authors explore their hypothesis of sex dependent responses to ALS and HDAC4 manipulation in parallel. I think this makes the report more complicated and it would be easier to follow if the authors first discussed their research about sex dependent responses to G93A, and then discussed their research about HDAC4
* The authors propose that the point of maximal weight is a superior measure of disease onset than measures that have previously been used to explore sex dependent differences in SOD1 mouse models. This is an interesting hypothesis and may have merit but I did not seen that the claim is really proven. Accordingly I was not convinced that this research really does clarify controversies from previous investigations of sex differences in SOD1 mice. (It certainly adds to the discussion but I would invite the authors to give opinion on this)
* I understood that the authors want to make some specific assertion about sex specific effects of SOD1 G93A overexpression on the kinetics of body weight changes. I find it difficult to fully interpret their data for this purpose without seeing data from WT mice. For instance, do wild type mice also show sex specific differences in body weight kinetics? Can it be excluded that these confound what the authors appear to imply is a sex dependent response to SOD1-G93A / ALS specifically?
* Figure 2b was not clear to me at first, probably because it was accompanied by 2 figures which showed time to events (the events being maximum weight or death). I think the content of fig 2b is already implicit in Fig 2 a/c so I would propose it could be dropped / supplemental
* It was not clear to me that there was a key novel point in section 2.3 / figure 3. Perhaps the authors can clarify the novelty/ need for a dedicated section? (or else move it to supplemental)
* Line 127 - "an almost significant difference between SOD1 and SOD1 127 HDAC4mKO females was found", I understand the thinking but more correct to say that no significant difference was found
Author Response
The authors explore 2 sets of research questions. The first concern whether there are sex dependent differences among SOD1-G93A transgenic mice concerning 1) the age at which maximum weight is achieved (pitched as a proxy for disease onset), 2) the rate of weight loss subsequent this timepoints and 3) total lifespan. The second set of questions concern whether there are sex specific differences in the phenotypic consequences of HDAC4 deletion in G93A mice. This question is relevant because HDAC4 inhibition has been proposed as a therapeutic strategy in ALS, but the authors have previously reported that HDAC4 deletion exacerbated negative outcomes in male G93A mice.
Sex specific effects are an important focus in ALS research, with several unresolved controversies in the literature. The authors do observe some phenotypic differences across sexes in their study, however I think the results could be presented a little more clearly and that certain data would perhaps fit better as supplemental materials. I think the extent of the conclusions that can be made for sex dependent differences of SOD1-G93A genotypes has some important limitations. I think the conclusion that the phenotypic impact HDAC4 manipulation in ALS contexts shows some variation across sex seems valid, but the authors could elaborate on why they feel this is important a little more in the discussion.
We thank the reviewer for the careful reading of the text and the deep understanding of the more critical points. We hope that our effort to address all the remarks succeeded in significantly improving the quality of the manuscript.
Additional specific comments:
* I found the title misleading. I read it as the authors would propose that HDAC4 was a mediator of sex dependent differences in the phenotype of G93A mice. Perhaps the take home message could be clearer
We have changed the title in: “Sex and HDAC4 differently affect the pathophysiology of Amyotrophic Lateral Sclerosis in SOD1-G93A mice”.
* The authors explore their hypothesis of sex dependent responses to ALS and HDAC4 manipulation in parallel. I think this makes the report more complicated and it would be easier to follow if the authors first discussed their research about sex dependent responses to G93A, and then discussed their research about HDAC4
The reviewer 1 raised a very similar concern. We do agree that an analytical, deconstructed, and ordered exposure of the data is very important. We did our best to separate the description of the sex-dependent and the genotype (HDAC4 KO)-dependent results, and also to discuss them accordingly. On the other hand, the added value of our work is indeed the interaction between the two variables, which must be taken into account as a whole. Therefore, it has not always been possible to keep the two separated. Below, the detailed response to the reviewer 1 on this issue, with examples etc.
In keeping with the need to focus on female-to-male comparison, we have separated the discussion of the data first by comparing females to males SOD1 mice, and then comparing the phenotypes in the absence or presence of HDAC4, as much as possible. For instance, see lines 84 (sex) and 89 (genotypes) for the body weight. To further help the legibility of the results, we referred in the text to statistic symbols when describing specific, significant differences (see line 99 as an example).
The statistical approaches we choose (2-way ANOVA, Cox etc.) actually take into account the two variables, i.e. sex and genotype, together by definition. For instance, when we described the significant effect on ALS onset for the two variables, i.e. sex (p=0.008) and genotype (p=0.002), by using the Cox regression test (see line 117), within the variable “sex” both genotypes are present, and vice versa. Based on all the above, the disjoint discussion of the sex differences and those due to the genotype is rarely possible.
Indeed, we think that considering the two variables together represents the major innovation and advance of our work; consequently, we chose not to change the analyses and statistics performed, but rather to implement the description of the results in the text.
* The authors propose that the point of maximal weight is a superior measure of disease onset than measures that have previously been used to explore sex dependent differences in SOD1 mouse models. This is an interesting hypothesis and may have merit but I did not seen that the claim is really proven. Accordingly I was not convinced that this research really does clarify controversies from previous investigations of sex differences in SOD1 mice. (It certainly adds to the discussion but I would invite the authors to give opinion on this)
Several papers presented a variety of ways to define ALS onset in SOD1 mice, including the following: a change in gait (Muscle Nerve. 2005 Feb;31(2):214-20. doi: 10.1002/mus.20255.), the decline in the number of motor neurons per histology section (Experimental Neurology 204 (2007) 260–263), the first out of three consecutive paw grip endurance test days in which a mouse failed to reach the cutoff time (J. Neurosci., August 24, 2005 • 25(34):7805–7812). In addition, several research groups adopted “the first sign of weight loss from peak body weight” as a measure of ALS onset (Theranostics. 2022; 12(12): 5389–5403. doi: 10.7150/thno.72614; Sci Adv. 2022 Jan 21;8(3):eabk2485. doi: 10.1126/sciadv.abk2485; Mol Ther. 2020 Apr 8;28(4):1177-1189. doi: 10.1016/j.ymthe.2020.01.005.). Noteworthy, “weight loss” has been considered a “well validated, robust, reproducible measure of ALS onset” in preclinical animal research in ALS in the “Guidelines for preclinical animal research in ALS/MND: A consensus meeting” (in Amyotrophic Lateral Sclerosis. 2010; 11: 38-45; DOI: 10.3109/17482960903545334). In our experience, this parameter is at the same time one of the most reproducible and reliable, also considering that this approach is poorly biased – the mouse body weight can be easily assessed by different persons over time, by using the same electronic balance. Therefore, we agree with this approach as a method for the pathology monitoring.
We believe that the ample discussion on sex differences, as well as body weight as an indicator of the course of the disease, clearly show the relevance of these issues and that the topic is (still) controversial. Our contribution may be of help for the discussion, even though it does not “clarify controversies” – we carefully avoided this statement.
* I understood that the authors want to make some specific assertion about sex specific effects of SOD1 G93A overexpression on the kinetics of body weight changes. I find it difficult to fully interpret their data for this purpose without seeing data from WT mice. For instance, do wild type mice also show sex specific differences in body weight kinetics? Can it be excluded that these confound what the authors appear to imply is a sex dependent response to SOD1-G93A / ALS specifically?
Sex-specific effects on mouse body weight are known indeed (https://www.criver.com/sites/default/files/resources/C57BL6MouseModelInformationSheet.pdf), and confirmed in this manuscript both in SOD1 and in SOD1 HDAC4mKO mice. The weight curve of our SOD 1 mice appears to be in the WT range for male and female mice, respectively, within the first 15 weeks of age; after that time point, the mice weight appears to keep growing with the same trend until the first signs of the disease onset. We agree that we cannot assert that the overexpression of the human mutant SOD1 gene specifically and differently affects body weight kinetics in mice. However, this is not the major point: all the mice shown in Figure 1 are SOD mutants and what we observe is an effect of time as well an effect of the genotype (HDAC4 KO) on their weight; we cannot tell if this adds up to a (potential) effect of the SOD1 G93A overexpression. We have better discussed this point in line 260 and 264 of the discussion.
* Figure 2b was not clear to me at first, probably because it was accompanied by 2 figures which showed time to events (the events being maximum weight or death). I think the content of fig 2b is already implicit in Fig 2 a/c so I would propose it could be dropped / supplemental
We agree with the reviewer’s opinion. We have moved panel 2b (ALS progression) in Supplementary Materials as Figure S1.
* It was not clear to me that there was a key novel point in section 2.3 / figure 3. Perhaps the authors can clarify the novelty/ need for a dedicated section? (or else move it to supplemental)
We agree that the former Figure 3 was not showing conceptually novel concepts, in need to be distinguished from Figure 2. Therefore, we have moved the two panels of figure 3 into the new figure 2, to complete the data about ALS onset and mouse survival.
* Line 127 - "an almost significant difference between SOD1 and SOD1 127 HDAC4mKO females was found", I understand the thinking but more correct to say that no significant difference was found
We have changed the text, accordingly, leaving the p value (p=0.096) in the text to imply the trend.

This manuscript is a resubmission of an earlier submission. The following is a list of the peer review reports and author responses from that submission.
Round 1
Reviewer 1 Report
In this report, the investigators show that the rate of weight loss is greater in both sexes in ALS mice where HDAC4 has been deleted in skeletal muscle (males > females), and a shortened survival in both males and females compared to just SOD1G93A alone. These patterns are consistent with the authors’ previous report of a protective effect of HDAC4 and are not unexpected since weight in the mouse models has long been recognized as a biomarker of disease progression. This reduces the novelty and significance of the report.
1. Combining all age populations, the ratio of females to males is closer to a 1.1/1.2 male to female ratio. Caucasian males with ALS are disproportionately high in the younger population as are African American females.
2. The methods section is very sparse and does not describe how some of the phenotype measures—particularly disease onset and survival—were measured. These are critical parameters for the paper. In the report the authors mention weight as a measurement of onset. It would seem that the males would then have an earlier onset based on their weight curves (Fig. 2A) but Fig. 2B shows the are overlapping with females.
3. In Fig 2 HDAC4 KO leads to shortened survival which was already reported by the authors. So the greater weight loss in these mutant mice and the pattern of males having shorter survival (which is consistent with the SJL hybrid mouse) is not particularly novel.
4. The findings of the authors in this paper and their prior report are conflicting with a report by Williams et al (20007902) who found that HDAC4 is inhibitory toward reinnervation and that miR-206, which suppresses HDAC4, mitigates the effect.
Reviewer 2 Report
In this manuscript the authors present follow up work from their recent interesting work relating to the negative effects of muscle HDAC4 deletion in male SOD1G93A mice on disease onset and NMJ/muscle innervation. The work herein adds further evidence to the effect of sex in ALS and related models emphasising the need for understanding sex-dependent phenomena in ALS disease pathology. With a predominant focus on weight changes in the SOD1G93A mouse model the authors find changes in weight and onset in females with HDAC4 deletion and draw relationships between weight loss and survival. The work represents a useful incremental advance on their prior publication whilst providing some new information on sex-dependent effects of muscular HDAC4 in an ALS context. I essentially recommend the paper for publication after clarification of the following:
1) In the sentence from line 55-60 there appears to an accidental omission after “… on ALS onset and…”. I’m guessing this is meant to say “HDAC4” where conflicting reports exist to suggest HDAC up-regulation may contribute or protect against disease progression in ALS.
2) As a suggestion, I recommend using “SOD1” as consistent nomenclature for specificity.
3) (input male HDAC4 data and figure suggestions + incorporation of un-normalized weight data to see comparative changes with time as a reference point).
4) In Figure 1 why were the SOD1 HDAC4mKO male data not included? These should be put in for a complete comparison.
5) The 12-week old body weight data in Figure 1a are stated as being “preonset” however, this isn’t necessarily true with regards to Figure 2b which shows a number of animals across different genotypes having registered disease onset by this time.
6) The paragraph section from lines 81 to 89 is confusing especially in the context of the data presented and referred to in Figure 1. It is stated that male SOD1 mice weighed more than female SOD1 mice, yet the bar graph shown would suggest no difference at all. My presumption is this refers to body weight of SOD1 males vs SOD1 females across multiple time points not just 12 weeks or 15 weeks (unless these two ages were selected together for such an analysis?), which would make sense with the use of two-way repeat measures ANOVA and reported significance. With this in mind, it would be best if the authors just simply put in unnormalized body weight changes over time for all genotypes to contrast with data shown in Figure 2a. This would be more appropriate for statistical interpretation using the two-way repeat measures ANOVA. It would also provide a clearer picture with un-normalised and normalised body weight shown as Figure 1a/b (to replace 12/15 week bar graphs) and subsequent figures for onset, progression, survival Kaplan-Meier curves and correlation analyses.
7) Line 111 states that SOD1 HDAC4mKO females weigh significantly more than compared to SOD1 HDAC4mKO males from week 16 to 18, however this normalised representation of the data can only be used to determine relative increases or drops in weight not actual body weight. The authors can only speak to SOD1 HDAC4mKO males showing a more rapid decline in weight compared to females over weeks 16 to 18. Discussion of actual body weight would require showing unnormalized body weight along with normalised body weight.
8) Line 118 talks about an almost significant effect for onset (p=0.05) based on sex. This suggests SOD1 and SOD1 HDAC4mKO males were combined and compared with SOD1 and SOD1 HDAC4mKO females for this test? If so, why was this done? It would be more informative to compare different ‘groups’ i.e., SOD1 HDAC4mKO males vs SOD1 HDAC4mKO females, and SOD1 males vs SOD1 females. SOD1 (male/female) vs SOD1 HDAC4mKO (male/female) can also be assessed this way with the conclusion seeming to be no onset difference between SOD1 HDAC4mKO males vs females, however an apparent earlier onset in SOD1 HDAC4mKO females vs SOD1 females in line with that seen for SOD1 HDAC4mKO males vs SOD1 males in the author’s prior work and in this paper (if statistically significant).
9) In line 126 it is said that lack of skeletal muscle HDAC4 affects survival with reference to female mice, whereas Figure 2c would suggest not change in survival for female SOD1 HDAC4mKO mice vs female SOD1 mice. I’m guessing the analyses done was combined male/female SOD1 HDAC4mKO mice vs combined male/female SOD1 mice? If so, as per comment above it would best to explicitly state this, and even better both show the combined sex Kaplan-Meier curves and run stats tests for individual ‘groups’ i.e., female SOD1 HDAC4mKO vs female SOD1; male SOD1 HDAC4mKO vs male SOD1; female vs male SOD1 HDAC4mKO; female vs male SOD1.
10) Whilst I appreciate the selection of weight loss in grams/day through 16-17 weeks as a means of deriving correlation with survival (as a more linear stage of weight decline due to disease), I feel conclusions drawn from the equation can be paradoxical. In the hypothetical of a male mouse dropping two grams in a day (say one of the male SOD1 HDAC4mKO or male SOD1 mice having shortest survival) this according to the equation would correspond to a survival in days of -2.42. I suggest sticking to the mention of the correlation, which makes sense, whereby a greater decrease in weight per day at that time point would rationally equate to shorter survival. The authors might also consider representing the correlation with individual mice as opposed to the ‘group’ median. This could be done in the context of one correlation line for all mice together and/or multiple correlation lines/analyses on one graph related to different groups (i.e., male SOD1, female SOD1, male SOD1 HDAC4mKO, female SOD1 HDAC4mKO) as this could be informative for the reader.
11) In lines 197-200 these statements would be aided by inclusion of the un-normalised body weight with age as suggested in previous comments.
12) Lines 214 and 215 state that SOD1 males lost body weight more rapidly and thus appear more prone to a severe form of disease. This would appear to imply that the loss of weight was a cause of earlier onset and decreased survival as opposed to a consequence. While the involvement of weight changes in ALS prognosis is has become established, I’m not this conclusion follows. The SOD1 males present with earlier onset compared to SOD1 females and so would be less inclined to weight gain over the period of time from 9 weeks when the data were normalised due to this average earlier onset.
13) The conclusion section states that HDAC4 affects survival in both male and female mice, however Figure 2c would appear to show no effect on survival with HDAC4mKO in female SOD1 mice (only when males and females are combined?). The conclusion needs to clearly reflect this point.
14) Some minor grammatical and spelling mistakes (including, but not limited to line 111 “significanly”, line 117 “a Kaplan-Meier curves”, line 131 “as apposed”, line 148, and line 149 “hgiher”.
Reviewer 3 Report
This study focuses on the importance of sex differences and disease kinetics in ALS disease using SOD1-G93A mouse model. Despite the current study arises an interesting fact about the importance of variability in sex in disease onset and progression, the protective role of HDAC4 in ALS is quite confusing and the data provided in this context is weak. The overall study is not novel and does not providing any mechanistic evidence by which HDAC4 works.